# Mucopolysaccharidoses: Cellular Consequences of Glycosaminoglycans Accumulation and Potential Targets

**DOI:** 10.3390/ijms24010477

**Published:** 2022-12-28

**Authors:** Andrés Felipe Leal, Eliana Benincore-Flórez, Estera Rintz, Angélica María Herreño-Pachón, Betul Celik, Yasuhiko Ago, Carlos Javier Alméciga-Díaz, Shunji Tomatsu

**Affiliations:** 1Institute for the Study of Inborn Errors of Metabolism, Faculty of Science, Pontificia Universidad Javeriana, Bogotá 110231, Colombia; 2Nemours Children’s Health, Wilmington, DE 19803, USA; 3Department of Molecular Biology, Faculty of Biology, University of Gdansk, 80-308 Gdansk, Poland; 4Faculty of Arts and Sciences, University of Delaware, Newark, DE 19716, USA; 5Department of Pediatrics, Graduate School of Medicine, Gifu University, Gifu 501-1193, Japan; 6Department of Pediatrics, Thomas Jefferson University, Philadelphia, PA 19144, USA

**Keywords:** endoplasmic reticulum, glycosaminoglycans, lysosome, mitochondria, mucopolysaccharidoses

## Abstract

Mucopolysaccharidoses (MPSs) constitute a heterogeneous group of lysosomal storage disorders characterized by the lysosomal accumulation of glycosaminoglycans (GAGs). Although lysosomal dysfunction is mainly affected, several cellular organelles such as mitochondria, endoplasmic reticulum, Golgi apparatus, and their related process are also impaired, leading to the activation of pathophysiological cascades. While supplying missing enzymes is the mainstream for the treatment of MPS, including enzyme replacement therapy (ERT), hematopoietic stem cell transplantation (HSCT), or gene therapy (GT), the use of modulators available to restore affected organelles for recovering cell homeostasis may be a simultaneous approach. This review summarizes the current knowledge about the cellular consequences of the lysosomal GAGs accumulation and discusses the use of potential modulators that can reestablish normal cell function beyond ERT-, HSCT-, or GT-based alternatives.

## 1. Introduction

The mucopolysaccharidoses (MPSs) are a group of rare genetic diseases caused by mutations in the genes encoding for the enzymes involved in the glycosaminoglycans (GAGs) catabolism, leading to their accumulation within the lysosome. GAGs are complex lineal carbohydrates composed of repeating disaccharide units of hexamine and uronic acid. Five GAGs have been described: heparan sulfate (HS), dermatan sulfate (DS), chondroitin sulfate (CS), keratan sulfate (KS), and hyaluronic acid (HA) [1].

The lysosomal accumulation of partially degraded GAGs was seen as the only cause of the pathophysiology in these diseases. Nevertheless, currently, it is well known that the pathophysiology of MPS results from the impairment of several cellular systems [2]. In fact, it has been described that GAG accumulation may trigger the storage of secondary substrates (e.g., glycosphingolipids, phospholipids, cholesterol, aggregated-prone proteins, and alpha-synuclein), inflammation, impairment of autophagy, changes in the intracellular trafficking of vesicles, mitochondrial dysfunction, oxidative stress, and dysregulation of signaling pathways [2,3,4,5,6].

Clinical therapeutic options include enzyme replacement therapy (ERT) and hematopoietic stem cell therapy (HSCT) for most MPS diseases [7]. Moreover, experimental therapies involving gene therapy [8,9,10], substrate reduction therapy, anti-inflammatory therapy, and pharmacological chaperone therapy [11,12,13] are under evaluation and could be potential alternatives shortly.

Approaches focusing on replacing missing enzymes are supposed to be the most promising alternatives; however, it has been shown that recovery of altered organelles and their related cellular processes positively impacts the whole cell homeostasis [2,14,15,16,17,18,19]. Thus, recognizing intracellular changes due to primary GAG accumulation is critical for attempting future holistic treatments. Herein, we review key aspects of GAGs, including the cellular consequences of their lysosomal accumulation and the potential use of modulators to recover the organelle/signal pathway affected.

## 2. Glycosaminoglycans: Structure, Biosynthesis, and Catabolism

GAGs are the most abundant biomolecules in the extracellular matrix (ECM). GAGs are a diverse class of long, linear, and heterogeneous polysaccharides characterized by disaccharide repeats composed of alternating units of uronic acid and amino sugar (Table 1), forming chains that range from 1 to 25,000 disaccharide units [20,21]. They are involved in critical biological processes, including ECM hydration, cell signaling, and regulation of growth factors [22]. Heterogeneity of the GAGs is based not only on the disaccharide composition but also on the polysaccharide chain length, the glycosidic linkage binding these units, either α or β, and the sulfation, acetylation, and epimerization patterns of the glycan chains. GAGs exist as free polysaccharide chains or are covalently linked to core proteins forming proteoglycans (PGs) [23].

### 2.1. Glycosaminoglycans Biosynthesis

The biosynthesis of GAGs (except for KS) begins in the cytoplasm with the formation of a tetrasaccharide linkage to the core proteins, catalyzed by the sequential actions of four glycosyltransferases, which adds one xylose, two galactose, and one glucuronic acid residues [24,25]. This tetrasaccharide block is a common precursor for the biosynthesis of HS and CS/DS polysaccharide chains, while three types of KS occur, depending on its core structure, and are detailed in the next section. These blocks are then transported into the Golgi lumen, where the polymerization occurs. The step that differentiates the formation of HS or CS/DS remains in the addition of the N-Acetylglucosamine (GlcNAc)- or N-Acetylgalactosamine (GalNAc)-units to the nonreducing end of the tetrasaccharide, respectively [24]. The polymerization continues in the case of HS by the alternating addition of GlcNAc and glucuronic acid (GlcA) residues by HS polymerases. In contrast, for CS/DS, the addition of GalNAc to the linker tetrasaccharide is required for the elongation [26]. After that, the backbones are subjected to modifications by epimerases or acetylases, increasing the structural heterogeneity of these molecules [27]. For KS, the only GAG composed of alternating β 1,3 and β 1,4 linkages between galactose and N-acetylglucosamine [28], biosynthesis begins with the building of the region binding with the core protein. Depending upon that linkage to the PG, three types of KS exist: KS I, found mainly in the cornea, is N-linked via asparagine in the core protein in a mannose-containing linkage oligosaccharide; KS II, frequently found in cartilage, is O-linked to serine/threonine via galactosamine; and KS III, mainly expressed in nervous tissue, is O-linked to serine/threonine via mannose and is the most sulfated of the three types [29].

#### 2.1.1. Keratan Sulfate

KS is the only GAG containing galactose (Gal) instead of uronic acids. Hence, it comprises alternating repeating units of β1,3-Gal and β1,4-GlcNAc [30]. Both structures may have 6-O sulfation. The molecular weight ranges from 6 KDa to >50 Kda; however, the degree of sulfation and size depends on the tissue from which it is found. For instance, KS I, found mainly in the cornea, is N-linked via asparagine in the core protein in a mannose-containing linkage oligosaccharide; KS II, frequently found in cartilage, is O-linked to serine/threonine via galactosamine; and KS III, mainly expressed in nervous tissue, is O-linked to serine/threonine via mannose and is the most sulfated of the three types [29].

#### 2.1.2. Heparan Sulfate (HS)

HS is synthesized as repeating disaccharides (GlcA and GlcNAc), suffering from N-deacetylations, N-sulfation, and epimerization of glucuronic to iduronic acid (IdoA). Both GlcA and IdoA can be sulfated at the C2 position, and glucosamine can be sulfated at the C3 and C6 positions [31]. The sulfation and acetylation patterns vary depending on the cell type. The molecular weight ranges from 5 to 50 Kda with 10 to 100 disaccharide units. HS is O-linked to serine residues of diverse core proteins, forming different HS proteoglycans (PG) types, including glypican and syndecans expressed on endothelial cell surface and leukocytes; perlecan, agrin, and collagen type XVIII are components of the subendothelial basement membrane [32]. Perlecan also regulates the development of blood vessels, cartilage, and endochondral ossification [33,34]. HS has the most ubiquitous expression and is the most diverse type of GAG, present in the mammalian cell surface of all tissues and the ECM. HS promotes hydration and participates in water and tissue homeostasis, organization of the ECM, and cellular migration; HS also interacts with collagen, growth factors, cytokines, chemokines, and interleukins, and hence plays a role in regulating inflammatory response, receptor-mediated signaling, phagocytosis, morphogenesis, and organogenesis [35,36]. HS storage disorders have been broadly correlated with central nervous system (CNS) impairment because of the involvement of HS in neuronal development as an essential component of the vascular basement membrane in the brain [37,38].

#### 2.1.3. Chondroitin (CS) and Dermatan Sulfate (DS)

CS and DS are synthesized as alternating linkages β1,3 and β1,4 between galactosamine and glucuronic acid (GalNAcβ1,4GlcAβ1,3)n in CS, or between galactosamine and mainly iduronic acid (GalNAcβ1,4IdoAβ1,3)n, in DS [26]. In CS, the GalNAc residues may be sulfated at C4 and/or C6, while GlcA can be sulfated at C2. Hence, based on sulfo group substitution and the type of uronic acid that each contains, multiple forms of CS have been described: CS-A is GlcA-GalNAc(4S); CS-B is also known as DS; CS-C is GlcA-GalNAc(6S); CS-D is GlcA(2S)-GalNAc(6S); CS-E is GlcA-GalNAc(4S,6S). Since the hybrid nature of DS structure is due to epimerization at C5 of GlcA to form IdoA, the sulfation at the C4 position is mainly found in DS. However, it can also be present at the C6 of GalNAc and C2 IdoA residues. The molecular weights of the CS/DS family of GAGs range from 2 to 50 kDa, and diverse biological functions have been described thanks to their high structural heterogeneity. CS/DS PGs are the major components of the CNS, participating in the axon growth tans synapse formation [39].

CS is one of the most critical constituents of bones due to its essential role in cartilage and other connective tissues. Moreover, it has been found to participate in the prevention of inflammation, immune modulation, maintenance of the structure and function of cartilage, and regulation of cell adhesion to the ECM [30]. For DS, the biological process described includes collagen organization, regulation of transforming growth factor (TGF)-β activity, stabilization of the basement membrane, and regulation of the cell–cell and cell–matrix interactions [26]. On the other hand, since DS shares structural similarities with lipopolysaccharide, DS activates the Toll-like receptor 4 (TLR4) signaling pathway, initiating the release of pro-inflammatory cytokines, and promoting the articular chondrocyte apoptosis. Decorin and biglycan are DS-/CS-PG that act as structural constituents of skin, cartilage, intervertebral discs, tendons, cornea, kidney, and muscle, promoting proper collagen fiber organization and endochondral ossification [40]. Biglycan is also present in bone and is vital in regulating skeletal growth by interacting with growth factors such as TGF-β, bone morphogenic protein 4 (BMP-4), and the Wnt signaling pathway [41].

### 2.2. Glycosaminoglycans Degradation

The stepwise degradation of the GAGs occurs within the lysosomes, where acid hydrolases degrade DS, HS, KS, and CS through the sequential removal of monosaccharides followed by the removal of sulfate groups, resulting in the complete degradation of the polysaccharide to its fundamental constituents’ components [42]. Misfunctioning of any of the 12 enzymes participating in the GAGs degradation results in eight different types of disorders known as MPS, characterized by the lysosomal accumulation of undegraded or partially degraded substrates in most cells, tissues, and organs, which ultimately results in multi-systemic failure and death (Table 2) [43,44].

Depending on GAG stored and the residual enzyme activity, MPS patients may present with hepatosplenomegaly, cardiac and pulmonary disease, skeletal deformities, earing and ophthalmological abnormalities, and central or peripheral nervous system impairment [45,46].

Although for many years, the pathogenesis and the clinical outcomes of the MPSs were considered a simple consequence of the cellular accumulation of GAGs. However, it is clear that GAGs are molecules biologically active, either free polysaccharide chains or attached to diverse proteins, playing essential roles in several biological activities, acting as modulators of proper organization of collagen fibrils and elastin fibers, immunomodulators, anticoagulants, antioxidants, anti-inflammatories, and neuroprotectors [47,48]. Consequently, any impairment in the degradation pathway can trigger intracellular pathological cascade activation.

## 3. Intracellular Organelle Impairment

In MPS, the lysosome and many other organelles are affected. It is well known that a global cell homeostasis impairment occurs, leading to the subsequent alteration at the tissue, organs, and systems levels. In the following sections, we will cover critical aspects of organelles impairment (summarized in Figure 1) and potential interventions for their recovery beyond classical approaches related to correcting the primary enzyme deficiency.

### 3.1. Lysosomal Impairment

Lysosomes process a broad range of materials as protein and nutrients products of endocytosis, phagocytosis, and autophagia, as well as sense environmental conditions to integrate a quick response through complexes signaling that allow homeostatic regulation, metabolic energy regulation, immune responses, and membrane sealing, among other regulatory pathways [2,49,50]. Because of their pivotal functions, GAG accumulation within the lysosome induces lysosomal stress, which is characterized by changes in lysosome biogenesis, increased intraluminal pH, membrane permeability, and crosstalk between organelles, leading to the activation of several pathways [49,50,51,52], with catastrophic consequences for the cell homeostasis. As stated by Simonaro, “*the lysosomal system shuts down, leading to abnormalities in endocytosis, autophagy, inflammation, and ultimately causing cell death*” [53].

Upon lysosomal GAG accumulation, the cell increases the lysosome biogenesis by activating the transcription factor EB (TFEB), a member of the microphthalmia-transcription factor E (MiT/TFE) transcription factor family. Members of MiT/TFE are responsible for the activation and regulation of several stress response pathways [49,51,52,54]. Dephosphorylation of TFEB by phosphatases calcineurin or protein phosphatase 2A (PP2A) allows a fast translocation to the nucleus to bind directly to the promoter region in the binding element called coordinated lysosomal expression and regulation (CLEAR), activating the expression of numerous genes involved in autophagy and lysosome biogenesis pathways [49,51,55,56]. Furthermore, an elevation of the calcium levels in the cytosol allows activation and translocation to the nucleus of TFEB by activating calcineurin [51,55].

Moreover, during lysosomal stress, an increased intraluminal pH is observed in MPS and other LSDs. The dramatic increase in the pH is caused by the dysregulation of membrane-resident ion channels or vacuolar-ATPases, given that the integrity of the membrane is affected by undegraded materials accumulated into the lysosome [2,51,57]. Since endolysosomes store ions like calcium and iron, their de-acidification induces efflux of those ions to the cytoplasm, impairing the activity of several organelles such as mitochondria and ER. Moreover, it has been described that de-acidification leads to changes in size and cytosolic mislocalization of endolysosomes [51].

Since the integrity of the lysosome membrane is affected by accumulating undegraded materials, such as GAGs, the lysosome releases several intraluminal molecules to the cytosol, such as galectins and cathepsins [49]. Galectins are a group of proteins involved in different physiological processes [49,58], and a role in regulating unfunctional lysosomes through autophagy has been reported [49] through the activation of AMP-activated protein kinase (AMPK), which results in the activation of the autophagy [49,59,60,61]. On the other hand, cathepsins are a group of proteases that participate in various physiological functions [62]. The secretion of cathepsins to the cytosol after lysosomal membrane impairment has been related to caspase-dependent and -independent cell death pathways [63].

### 3.2. Autophagy Impairment

Autophagy is an essential evolutionary cellular process that degrades cytoplasm components assisted by the lysosome [18,64]. Classically, three autophagy classes have been described, chaperone-mediated autophagy, microautophagy, and macroautophagy. Although these types of autophagy differ in the stepped mechanism to recognize and catch the cytoplasm cargo, all of them deliver the cargo to the lysosome for their degradation. A complete review of molecular mechanisms in autophagy was previously published by Pierzynowska et al. (2018) [18], and Figure 2 shows the molecular mechanisms of the autophagy process.

Since lysosomal fusion is critical to degrade the cargo inside autophagosomes, during MPS, the autophagosome–lysosome (AL) pathway is also affected [2,65,66]. Early reports showed that autophagosomes are accumulated because of the impaired AL fusion in MPS IIIA and multiple sulfatase deficiency [67], emphasizing the close relationship between the lysosome and autophagic pathways. In concordance, LC3 increased in the cerebral cortex and stratum from MPS IIIB [68]. Interestingly, by treating animals with CLR01, a molecular tweezer, significant neuroinflammatory findings such as astrogliosis and microgliosis were significantly decreased in untreated animals, with improvement in memory tests [68]. Elevated LC3-II levels were also reported in MPS IIIC mice [69].

Previously, Maeda et al. (2019) found abnormal autophagy in MPS II mice [70]. Immunostaining of p62 showed an increase over time in MPS II mice, compared to wild type, as well as in the cytoplasm vacuoles in neurons, microglia, and pericytes, suggesting a lack of AL fusion. Remarkably, by using chloroquine, an autophagy-suppressing drug [71], neurodegeneration was prevented, while microglia and pericytes remained vacuolated [70].

Increased phosphorylation levels on P70S6K and ULK1 were reported in MPS II, VI, and VII chondrocytes. Furthermore, MPS VII chondrocytes have shown nuclear localization of TFEB and TFE3, suggesting an mTOR-independent nuclear accumulation, although the autophagic vacuole biogenesis is not primarily affected. Autophagosome maturation was significantly involved in MPS VII chondrocytes [72]. For MPS IVA, a significant decrease in p62 and LC3B was found in MPS IVA patients, suggesting a blocked autophagosome formation [73,74].

Although AL impairment has been widely recognized in MPS, some reports contrast those findings. For example, LC3-I conversion to LC3-II was found unaltered in MPS IIIB mouse cortical neurons, suggesting that macroautophagy was unaffected in this model [75]. Similar findings were reported in peritoneal MPS I mouse fibroblasts [76].

**Figure 1 ijms-24-00477-f001:**
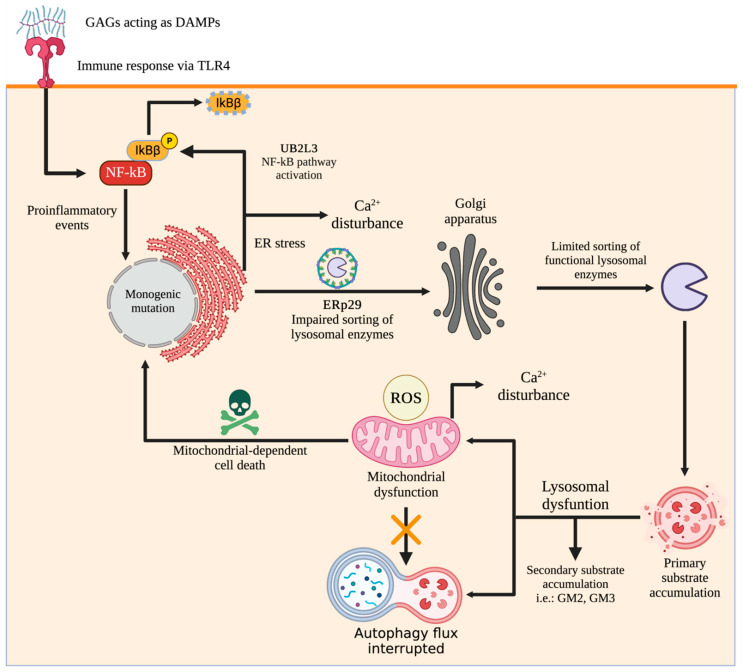
Global cellular changes occur during GAG accumulation. Note that primary GAG accumulation affects not only lysosomal function but also any intracellular organelles contributing to altered cell homeostasis. Since GAGs act in cell–cell communication, their extracellular accumulation, due to lack of plasma turnover [77], promotes inflammation-related activation pathways since they are recognized as damage-associated molecular patterns (DAMPs). For instance, soluble HS fragments can act as Toll-like receptor 4 antagonists supporting the pro-inflammatory events of MPS III [78,79]. This figure was created with BioRender.com.

### 3.3. Mitochondrial Impairment

Primary lysosomal dysfunction occurring during MPS can affect mitochondrial functions. Mitochondria is the primary energy producer in eukaryotes through oxidative phosphorylation (OXPHOS). It is also a critical organelle involved in vital cellular processes such as calcium homeostasis, lipid metabolism, viral response, and programmed cell death [80]. OXPHOS is a predominant source of reactive oxygen species (ROS). Therefore, mitochondrial impairment results in imbalanced ROS levels [81]. ROS are oxygen-containing molecules with a high affinity to react with macromolecules, including DNA, proteins, and lipids [82].

**Figure 2 ijms-24-00477-f002:**
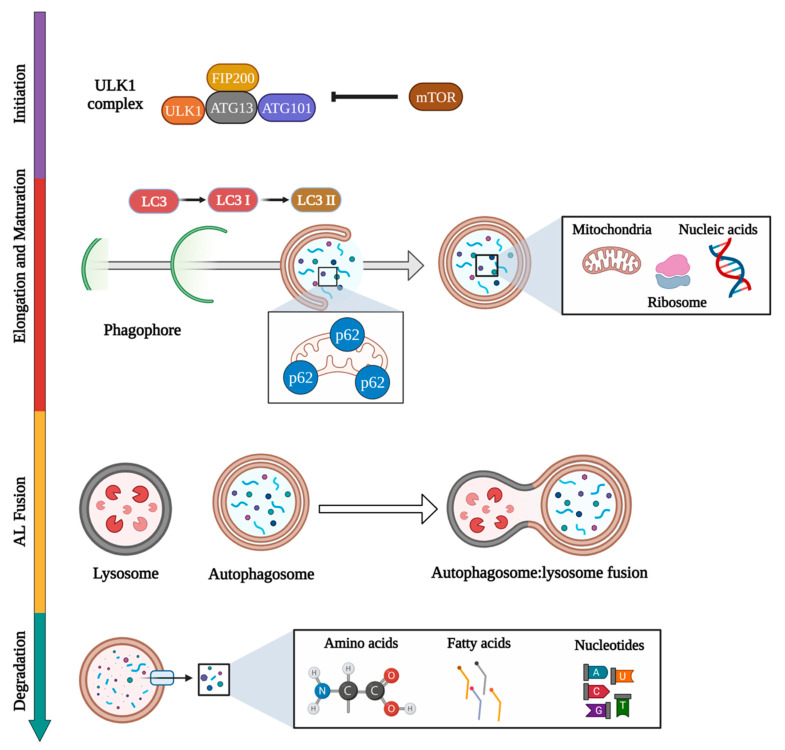
Schematic representation of the autophagy pathway. After extra- or intra-cellular autophagy stimulus such as nutrient starvation or oxidative stress [83], autophagy is triggered by the interaction of several effectors such as ULK1, ATG13, FIP200, and ATG101. In a nutrient-rich environment, the mTOR kinase hyperphosphorylates ATG13, avoiding binding to ULK1 and FIP200 [83]. During elongation and maturation, proteins and lipids are recruited to the phagophore biogenesis. In this process, several steps take place, such as the proteolytic cleavage of LC3 to LC3 I and its later conjugation with phosphatidylethanolamine to form LC3 II. LC3 II arrives at the autophagosome membrane, which has selection cargo engulfment functions [84]. Cytosolic structures that will be degraded via AL undergoes p62 labeling. According to the cargo, autophagy can be termed mitophagy or ribophagy [85]. Then, the autophagosome is fused to the lysosome in a SNARE-mediated process [86]. Several proteins, such as LAMP-2, Rubicon, and monomeric GTP-ases, contribute to the AL fusion [18]. Finally, the acid hydrolases inside the lysosome are released to the phagosome to digest the loaded cytosolic components. New macromolecules such as amino acids, fatty and nucleic acids are now available for cellular needs [18,83]. This figure was created with BioRender.com.

Recent studies have shown the critical role of ROS in the pathophysiology of MPS. For MPS II, oxidative stress has been involved in the pathogenesis. In this regard, Filippon et al. (2011) tested 12 MPS II patients aged from one to seven years old concerning oxidative stress before and during ERT treatment [87]. Blood samples were collected from MPS II patients before starting treatment and during each session of ERT. The results revealed clearly that malondialdehyde (a biomarker of lipid peroxidation), carbonyl groups, and erythrocyte catalase activity were higher in plasma before ERT than in the healthy control group. Moreover, total antioxidant levels and levels of the protein-bound sulfhydryl group were lower than control groups before ERT, suggesting an imbalance between anti- and pro-oxidant molecules in MPS II. Despite ERT leading to a decrease in the malondialdehyde levels, an increase in the sulfhydryl groups was observed compared to findings before treatment. Furthermore, ERT does not significantly impact plasma antioxidant levels, carbonyl groups, catalase, or superoxide dismutase activities, suggesting a limited effect of the ERT on the pro-oxidant profile observed in MPS II patients [87].

Oxidative stress was also evaluated in MPS IIIA mouse cerebral tissues in an early study conducted by Arfi et al. (2011) [88]. For instance, oxidative stress-related genes such as the inducible nitric oxide synthase (fold change: 2.1), NADPH oxidase subunit (fold change: 2.6), superoxide dismutase 2 (fold change: 1.9), glutathione peroxidase (fold change: 2.1) were found overexpressed in the brain of MPS IIIA mice, compared with healthy animals [88], supporting the link between a pro-oxidative profile and the CNS impairment in the MPS IIIA. Interestingly, by using an antioxidant treatment based on vitamin C administration, animals failed to show any improvement in oxidant profile [88], contrary to other LSDs, such as Sandhoff (SD), where the same treatment led to an improvement in the pro-oxidant findings in SD mice [89].

In MPS IVA, the analysis of blood and urine samples from patients showed lower levels of defensive antioxidants, such as glutathione, than healthy control groups, suggesting an oxidative homeostasis impairment [90]. Furthermore, superoxide dismutase levels increased while glutathione levels decreased in red blood cells [90,91]. Free radicals may oxidize lipids and proteins, resulting in isoprostanes and dityrosine yields, which can be used as biomarkers of oxidative stress [92,93]. Increased isoprostane and dityrosine levels in urine samples from MPS IVA patients were reported, which suggested lipid and protein damage, respectively [90].

### 3.4. Endoplasmic Reticulum Stress

Since most mutations in MPS can induce misfolding of lysosomal enzymes [13], ER stress can also be present. For example, by using transcriptomic approaches, it has been found that IER3IP1, a gene encoding for a resident ER membrane protein, is significantly downregulated in almost all MPSs except in MPSs IIID and VI, concerning unaffected human dermal fibroblast [94]. Although IER3IP1 has been associated with normal brain development [95], which could explain the symptoms of neurodegenerative MPS, it is well known that some of them, like MPS IVA, do not elicit such impairment [96,97]. In this sense, new approaches have attempted to elucidate IER3IP1′s functions. For instance, it was observed that IER3IP1 decreases the unfolding protein response (UPR) by shutting down two key activators, IR1α and PERK, and preventing caspase-mediated cell death [98].

These results suggest a new role for IER3IP1 in the context of MPS, at least when mutation cause misfolding. In this scenario, Osaki et al. (2019) recently demonstrated that attenuated (A85T)- and severe (R468Q)-associated IDS mutations are attached by calnexin (CNX) to perform refolding inside ER [99]. Interestingly, the authors reported an increase in the IDS band in Hela cells expressing mutated IDS (R468Q) upon siRNA transfection against HDR1, an E3 ligase involved in ER-associated degradation (ERAD) [100], suggesting that mutant IDS is degraded by the ERAD pathway because of the unsuccessful refolding process by CNX. In contrast, A85T seems susceptible to CNX refolding, and the later IDS sorting to the lysosome could exert its canonical function [100]. Furthermore, it was previously shown that knocking down HDR1 and ERdj3 (an HSP40 co-chaperone family member) leads to an increase in the translocation ratio of mutant IDS carrying out A85T to the lysosome [101,102], evidencing potential therapeutic targets for MPS II [103]. ER stress was also recognized by the impairment of endoplasmic calcium concentration in MPS I mice, although no apoptosis-related cell death was observed [76].

In concordance, proteomic analysis has shown that UB2L3 and ERP29 remain downregulated in MPS IVA leukocytes [104]. UB2L3 is a ubiquitin-conjugating E2 enzyme promoting the translocation of NF-κB to the nucleus when IL-1 and TNF-receptors are activated, suggesting a role in the pro-inflammatory events which take place in MPS IVA [90]. Moreover, UB2L3 has been found in the glycogen synthase kinase 3b (GSK3β)/p65 signaling pathway by decreasing the p65 levels and the apoptosis ratio [105]. Since p65 is part of the NF-κB structure [106], a direct relation between UB2L3 and the NF-κB regulation by the GSK3β ubiquitin-dependent degradation process was established [105]. On the other hand, ERP29 is a well-characterized UPR-related protein that facilitates the transport of proteins from ER to the Golgi apparatus [107]. Its dysregulation could be associated with impairment traffic of the GALNS enzyme between both organelles.

### 3.5. Apoptosis Activation

Despite several cell death mechanisms described in physiological and pathological scenarios [108], apoptosis is the most documented in MPS. Apoptosis can occur by several stimuli coming from intra- and extracellular environments, and it seems to be the last cellular decision under irreversible vital pathways dysregulation [109]. One is the calcium homeostasis imbalance which can trigger intrinsic apoptotic cells by accumulating inside the mitochondria. Calcium is primarily stored in the ER, where resident antiapoptotic proteins such as Bcl-2 control its ratio, decreasing the potential risk of harm upon calcium release to the cytoplasm [110]. Although this relationship was observed clearly for splenic lymphocytes and macrophages from MPS I mice in early studies conducted by Pereira et al., 2010 [111] and Viana et al., 2017 [76], later reports showed the opposite behavior in fibroblasts. MPS I fibroblasts seem more resistant to staurosporine-mediated cell death than normal cells [76], suggesting that apoptosis activation could be cell-specific.

Apoptosis was also identified as a critical finding in neuronal cells from MPS II-specific induced pluripotent stem cells (iPSC) [112], MPS III [88,113,114], MPS IV [115], and MPS VI rats [116,117,118]. However, dissecting mechanisms remain to be elucidated.

### 3.6. Immune Response Activation

Accumulating GAGs and secondary storage materials trigger innate immune responses via the TLR4 pathway, leading to inflammation in various organs [119,120,121,122,123]. The most detailed molecular pathway of how GAGs excite inflammation has been elucidated for HS. Many chemokines (e.g., IL-1) utilize HS as a coreceptor for binding with cognate ligands; the sulfation state of HS is essential for regulating this interaction [20,121,124]. Some of these chemokines appear to bind preferentially to the 2-O sulfated site of uronic acid, which constitutes HS. Among these chemokines is stromal cell-derived factor-1 (SDF-1 or CXCL12), associated with calcium release and chemotaxis of neutrophils, monocytes, and T lymphocytes [125,126]. In addition, interleukin 8, a chemokine known as a neutrophil chemotactic factor, also prefers 2-O sulfated sites [127]. The chain length of HS is also crucial for facilitating chemokine binding to HS. Moreover, it has been shown that at least a hexameric to a 20-mer disaccharide is required to promote binding [20]. The binding of chemokines to HS chains is essential in immune responses because some chemokines are activated by attachment to HS chains to form oligomers on those chains [128,129].

In addition, HS appears to act as a DAMP able to bind TLR4 directly and work as an agonist to stimulate this receptor [130,131]. Brennan et al. (2012) showed that HS activates TLR4 on dendritic cells in vitro, causing accelerated maturation of dendritic cells and alloreactive T-cell responses [130]. They also performed in vivo experiments in allogeneic HSCT in mice. They showed a positive correlation between serum HS concentration and the severity of graft-versus-host disease (GVHD), suggesting a strong association between HS and immune responses [130]. Goodall et al. (2014) demonstrated that soluble HS induces the release of pro-inflammatory cytokines from human peripheral blood mononuclear cells and mouse splenocytes in vitro [131]. Furthermore, treatment of splenocytes from MyD88 or TLR4-deficient mice with β-D-endoglucuronidase heparanase (HPSE) to increase the concentration of soluble HS revealed that a TLR4-mediated pathway is involved in this phenomenon [131].

HS binds with many extracellular proteins, not only cytokines and chemokines, but also matrix components, enzymes, coagulation factors, complements, growth factors, and morphogens [129,132,133]. Because the sulfated domain of HS typically allows for growth factor binding, excessive or abnormal sulfation can prevent or even dysregulate HS-dependent processes like neuronal proliferation and survival, synapse formation, and maintenance of function [125,134]. An early study on MPS I showed that HS accumulation upregulates HS sulfation via positive feedback [135]. This positive feedback of sulfation may further exacerbate neurologic symptoms through increased binding of HS to growth factors and excessive immune activation. In fact, highly sulfated disaccharides have been recognized as potential endogenous DAMPs, and it has been hypothesized that they could activate TLR-4 similar to the mechanism described for bacteria [78].

As mentioned above, activation of innate immunity is deeply related to the symptoms and progression of MPS. However, the relationship between adaptive immunity and MPS signs remains unclear in clinical settings. The lysosome pathways should be more affected. In adaptive immunity, lysosomes are essential organelles for professional antigen-presenting cells because antigens taken up by these cells need to be processed by lysosomes before they can present epitopes via Class II major histocompatibility complex (MHC) molecules [135], which are critical for CD4+ helper T cells to recognize the invasion of pathogenic microbes. Using an MPS VII mouse model, Daly et al. (2000) demonstrated a blunted T cell proliferative response to protein antigens and decreased antibody production [136].

Interestingly, similar responses were not seen in peptide antigens [136], which supports the inference that proteolysis in lysosomes is impaired in MPS. On the other hand, Class I MHC molecules do not require lysosomes. Instead, they need proteolysis of the antigen by proteasomes in the cytoplasm [137]. DiRosario et al. (2009) found elevated expression of CD8+, but not CD4+, using gene expression microarrays and real-time quantitative reverse transcriptase-polymerase chain reaction with brains from MPS IIIB model mice [138]. Class I MHC antigens were also upregulated, but not Class II [138]. In conclusion, humoral immunity is affected to a greater extent than cell-mediated immunity in patients with MPS.

## 4. Modulation of Altered Intracellular Pathways as a Potential Therapeutic Approach

Lysosomal enzymes contain mannose 6-phosphate (M6P) residues, which are recognized by the mannose 6-phosphate receptor (M6PR) and transported into the lysosome (Figure 3). The M6PR receptors on the cell membrane recognize lysosomal enzymes outside the cell, resulting in the internalization of the enzyme into the lysosome in a cross-correction mechanism [9]. Cross-correction is the rationale for classical strategies focused on recovering enzymes, such as ERT [139,140], HSCT [97,141,142], or GT [12,143,144], and are well reviewed elsewhere.

### 4.1. Substrate Reduction Therapy (SRT)

SRT aims to limit substrate synthesis by interfering with metabolic pathways related to GAG biogenesis [145]. For instance, small molecules such as genistein, a non-toxic isoflavone, induce the GAG reduction in many organs with behavioral correction in MPS II and IIIB mice [146,147]. Genistein acts by impeding the expression of genes encoding GAG synthetases, which leads to a decrease in the GAG levels in cells through negative regulation of epidermal growth factor receptor (EGFR) [148,149]. Additionally, genistein has been described as an mTOR inhibitor inducing dephosphorylation of TFEB and its translocation to the nucleus, where it positively regulates the expression of lysosomal biogenesis-related genes, ultimately leading to lysosomal degradation of GAGs [54]. A study of MPS II patients treated with genistein thought 26 weeks improved joint mobility [150]. Even with the promising results, clinical trials have not confirmed genistein’s pre-clinical success [151]. Another inhibitor of GAG biosynthesis is rhodamine B (RhoB), which was assessed in MPS IIIA mice with a significant reduction toward normal levels upon treatment with RhoB [152]. A slight improvement in urinary GAGs levels and cognitive tests were also reported for MPS I mice after treatment with RhoB [153].

Synthetic xylosides such as 5-thio-β-d-xyloside odiparcil, which compete with endogenous xyloside-containing core proteins for GAG assembly [154], result in the excretion of GAGs through urine instead of classical lysosomal accumulation [155]. Odiparcil is an orally active compound that allows the synthesis of soluble GAGs, mainly CS and DS. The neosynthesized solubles GAGs are then excreted in urine. By diverting endogenous GAG synthesis to the synthesis of soluble odiparcil-linked GAGs, odiparcil should decrease the intracellular pool of GAGs and consequently decrease the lysosomal GAG accumulation. A clinical trial of odiparcil demonstrated it to be safe in MPS VI patients [156], suggesting a novel treatment not only for MPS VI but also for other types of MPS.

### 4.2. Autophagy Recovery

As discussed in Section 3.2, autophagy can be impaired in some MPSs because of lysosomal homeostasis disruption; thus, stimulation of autophagy might be a potential target for treating patients. Autophagy modulators such as resveratrol and genistein, which were reviewed by Rintz et al. (2020) [19] and Pierzynowska et al. (2018) [18], respectively, which are recognized to restore autophagy flux and could be included in the management of MPS patients. Likewise, chloroquine, an autophagy inhibitor, was administered orally to MPS II mice. Chloroquine reduced neuronal vacuolation and eliminated neuronal cells with abnormal inclusions without suppressing microglia and pericyte damage [70]. Similarly, fluoxetine, an autophagy promoter, improves the function of lysosomes and activates autophagy through TFEB activation in a RagC-dependent mechanism [16]. In embryonic MPS IIIA mice fibroblasts, fluoxetine reduced HS accumulation through increasing lysosomal exocytosis. Likewise, MPS IIIA mice treated with fluoxetine showed a decrease in GAG and aggregated autophagic substrates accumulation, which led to inflammation and improved behavior [16].

### 4.3. Mitochondria and Oxidative Stress

Reduction of mitochondrial impairment and oxidative species appears to be one of the potential treatment options for MPS patients [157]. Coenzyme Q10 (CoQ) is a lipid that acts in the mitochondrial respiratory chain as an electron transporter essential for adenosine triphosphate (ATP) synthesis and a lipophilic antioxidant, among other functions. Secondary CoQ10 deficiencies result from gene mutations unrelated to the CoQ10 biosynthetic pathway or non-genetic factors. Secondary CoQ10 deficiencies have been linked with many disorders, including primary mitochondrial respiratory chain disorder, cardiovascular disease, chronic kidney disease, type II diabetes, and metabolic syndrome. The presence of low plasma CoQ value was reported in MPS and phenylketonuria [156]. A deficit of CoQ10 status has been reported in MPS III A and B patients [158]. However, there was no detectable impairment in the CoQ10 biosynthetic pathway. Evidence for secondary CoQ10 deficiencies has also been reported in association with HMG-CoA reductase inhibitors, ‘statins’, and Amitriptyline therapy. Treatment with CoQ10 in non-MPS LSDs mouse models reduced the level of ROS and cell death and restored mitochondrial membrane potential [159]. CoQ10 was decreased significantly in plasma from MPS patients compared with healthy individuals [160]; thus, implementing CoQ10 as a treatment could have a positive outcome.

Additionally, modulation of sirtuin activation, such as sirtuin 1 (SIRT1), could be another strategy for restoring global cell homeostasis. For instance, SIRT1 increases mitochondrial biogenesis through PGC1α transcription factor activation [161]. Treatment with resveratrol activates SIRT1 and increases the lifespan of mouse models of LSD (infantile neuronal ceroid lipofuscinosis) through mitochondrial function recovery [157] and could be attempted in several MPS models.

### 4.4. Immune Response Modulation

Due to substantial inflammation induced by cell pathology progress over MPS, several immune modulators have been investigated in both clinical and preclinical stages [162,163]. So far, immune modulators have only been studied in combination with ERT [14,118].

As mentioned above (Section 3.6), the TLR4 receptor is overly stimulated in MPS. Thus, increased inflammation, mainly by TNF-α, appears in MPS patients. Promising strategies to target this mechanism have been tested, including adalimumab, a monoclonal antibody that inhibits TNF-α. Adalimumab was confirmed to be safe and effective (improve pain and physical and neurogenic function) in the clinical trial for both MPS I and II (NCT02437253; NCT03153319) [163]. Another monoclonal antibody—alemtuzumab—targeting CD52 resulted in a profound but transient peripheral immune depletion [164]. Alemtuzumab was tested in a clinical trial for different types of MPS patients I and II as a polylactic regimen before HSCT [14]. Although cognitive skills were improved, a mild developmental delay was also noticed [165].

Due to their size, most cytokines produced in the periphery can cross the blood-brain barrier (BBB), affecting both CNS and systematic disease during MPS. Thus, lowering peripheral inflammation would improve the neurological outcomes of the disease [14]. Even if therapeutical antibodies do not cross BBB, their use in the combined treatment context might secondarily improve CNS inflammation. Abatacept, which blocks the co-stimulatory signal mediated by CD28–CD80/86 engagement, which is required for T-cell activation, is currently tested in children undergoing unrelated HSC transplant for serious non-malignant diseases—MPS I patients will be recruited as well (NCT01917708). It is administered in combination with cyclosporine and mycophenolate mofetil as GVHD prophylaxis. Tolerability and immunological effects will be assessed in the trial.

Pentosan polysulfate (PPS) is a well-known molecule that also suppresses the activation of TLR4 in in vitro and in vivo MPS models. For instance, decreased neuroinflammation and neurodegeneration in the MPS IIIA mouse brain [166] and enhanced delivery to the bone in the MPS VI mouse model [15,167] are observed upon PPS administration. PPS reached clinical trials with MPS I patients. PPS was well tolerated and led to a decrease in urine GAG levels and bone pathology improvement [168]. Likewise, MPS II patients treated with PPS showed decreased levels of TNF-α [162], supporting its use as a potential immune modulator.

Furthermore, GAG accumulation, impaired autophagy, mitochondrial dysfunction, and oxidative stress ultimately lead to NLRP3 inflammasome activation with IL-1β release [78]. Anakinra, a human antagonist of IL-1 receptor (IL-1R), is currently being tested in a clinical trial for MPS III patients (NCT04018755). Anakinra can cross BBB [169], which may increase its therapeutic efficacy for CNS symptoms in MPS patients.

## 5. Perspectives

During decades, MPS pathophysiology was only associated to the GAGs built-up within the lysosome. Nevertheless, in recent years, a significant body of evidence has shown that several cellular pathways are also affected in these disorders. In this sense, new treatment alternatives, such as stand-alone or co-administration, could be implemented. For instance, while the existing therapies only focus on the specific deficient enzyme supply, the use of modulators to restore affected organelles for recovering cell homeostasis is an alternative therapeutic approach to several MPS. Cell-homeostatic modulators as small molecules may have several advantages, including: (1) applicable to different types of MPS (broad application), (2) less expensive compared to ERT (low cost), (3) the possibility of oral administration and/or existing drug (feasibility), and (4) impact to hard-to-reach tissues including cornea, bone, brain, and/or heart valves as a small molecule (high penetration). The design of potential modulators of the ER–Golgi–lysosome pathway should be a major research focus in the near future. Through the recovery of this critical pathway, pathological events such as aberrant immune response activation could be mitigated. The use of combined approaches, based on organelles modulators and enzyme supply strategies, may positively impact the natural history of systemic diseases like MPS.

## Figures and Tables

**Figure 3 ijms-24-00477-f003:**
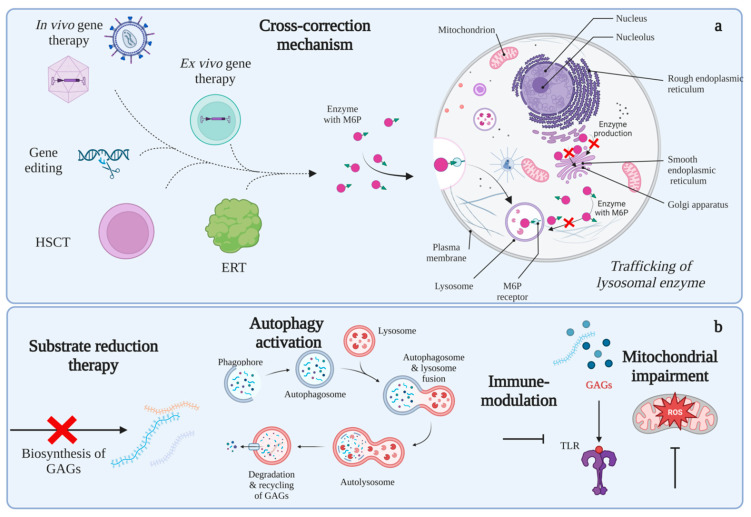
Potential therapies for MPS. Panel (**a**) shows possible treatments based on the cross-correction mechanism, including gene therapy, ERT, and HSCT. Panel (**b**) shows potential therapies not based on cross-correction mechanisms, such as substrate reduction therapy, autophagy activation, immune modulation, and mitochondrial impairment. Note that every organelle affected in MPS can be susceptible to intervention to recover cell homeostasis, together with current strategies to supply missing enzymes. This figure was created with BioRender.com.

**Table 1 ijms-24-00477-t001:** GAGs structures.

GAG	Monosaccharides Constituents	Chemical Structure
Keratan sulfate	Galactose N-acetylglucosamine	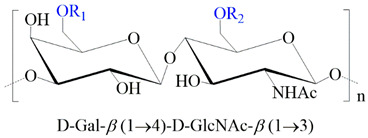
Heparan sulfate	Glucuronic acid N-acetylglucosamine	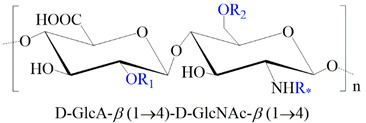
Chondroitin sulfate	Glucuronic acid N-acetylgalactosamine	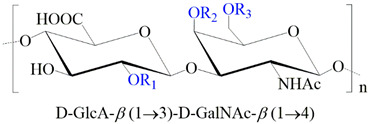
Dermatan sulfate	Iduronic acid N-acetylgalactosamine	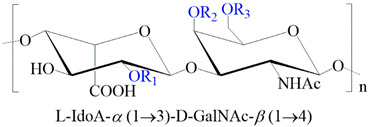
Hyaluronan	Glucuronic acid N-acetylglucosamine	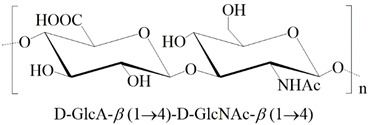

R_1,2,3_: -H or -SO_3_^−^. R_*_: -H, -SO_3_^−^, or -COCH_3_.

**Table 2 ijms-24-00477-t002:** Classification of MPS and major characteristics.

Type (Eponym)	OMIM	Affected Gene	Clinical Characteristics	GAGs
MPS I (Hurler)	607014 (Hurler)	*IDUA*	Corneal clouding, multiple dysostoses, organomegaly, cardiac disease, CNS impairment	DS and HS
	607015 (Scheie)		Corneal clouding, joint stiffness, normal CNS	DS and HS
	607016 (Hurler-Scheie)		Intermediate phenotype between Hurler and Scheie phenotypes	DS and HS
MPS II (Hunter)	309900	*IDS*	Multiple dysostoses, organomegaly, CNS impairment	DS and HS
MPS IIIA (Sanfilippo A)	252900	*SGSH*	CNS impairment, hyperactivity, mild somatic manifestations	HS
MPS IIIB (Sanfilippo B)	252920	*NAGLU*	Like MPS IIIA	HS
MPS III C (Sanfilippo C)	252930	*HGSNAT*	Like MPS IIIA	HS
MPS IIID (Sanfilippo D)	252940	*GNS*	Like MPS IIIA	HS
MPS IVA (Morquio A)	253000	*GALNS*	Short stature, skeletal dysplasia, corneal	KS and CS
MPS IVB (Morquio B)	253010	*GLB1*	Like MPS IVA	KS
MPS VI (Maroteaux-Lamy)	253200	*ARSB*	Multiple dysostoses, corneal clouding, cardiac disease	DS
MPS VII (Sly)	253220	*GUSB*	Multiple dysostoses, hepatosplenomegaly, mild to severe central nervous system involvement	HS, DS, and CS
MPS IX (Natowicz)	601492	*HYAL1*	Short stature, soft-tissue masses	HA
MPS X	619698	*ARSK*	Short stature, coarse facial features, dysostosis multiplex, cardiac and ophthalmological abnormalities.	DS

IDUA: α-L-iduronidase. IDS: Iduronate-2-sulfatase. SGSH: Heparan N-sulfatase (sulfamidase). NAGLU: α-N-acetylglucosaminidase. HGSNAT: Acetyl CoA: α-glucosamine N-acetyl transferase. GNS: N-acetylglucosamine-sulfate-6-sulfatase. GALNS: N-acetylgalactosamine-6-sulfate sulfatase. GLB1: β-galactosidase. ARSB: N-acetylgalactosamine-4-sulfatase (arylsulfatase B). GUSB: β -glucuronidase. HYAL1: Hyaluronidase. ARSK: Arylsulfatase K. DS: Dermatan sulfate. HS: Heparan sulfate. KS: Keratan sulfate. CS: Chondroitin sulfate. HA: Hyaluronic acid.

## Data Availability

Not applicable.

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
