# Peer review of "Mucopolysaccharidoses: Cellular Consequences of Glycosaminoglycans Accumulation and Potential Targets"

_ijms, 2022, doi:10.3390/ijms24010477_

Round 1
Reviewer 1 Report
The review by Leal et al aims to summarize the findings regarding mucopolysaccharidoses (MPS). This review is structured into chapters: introduction, Glycosaminoglycans: Structure, biosynthesis, and catabolism, Intracellular organelle impairment, Modulation of altered intracellular pathways as a potential therapeutic approach and Perspectives.
The review is readable, understandable and introduces the reader to the subject of MPS.
Unfortunately, there are a few inaccuracies or errors in the thesis that still need to be corrected:
Lines 61-64 you wrote: “They are involved in 62 critical biological processes, including ECM hydration, cell signaling, and regulation of 63 growth factors (Table 1)” but in Table 1 there are only nemae of GAG, Monosaccharides constituents and chemical structure! It is the same with the description of Table 1: "Table 1. GAGs structure and their biological functions”
Please thoroughly review the text referencing the table and unify the text and content of the table.
Line 159: please, move the description of Table 2 to a new page above the table.
Line 518 you wrote: ” As mentioned above (section 4.6)…” However, this review doesn't have section 4.6! Please carefully read this text and correct the links to the other sections.
Finally, the weakest part I find section 5 Perspectives. Please try to improve this section with your findings and what would be key to focus research on in the near future. Or where you see the most potential in research.
Author Response
Reviewer 1
Comments and Suggestions for Authors
The review by Leal et al. aims to summarize the findings regarding mucopolysaccharidoses (MPS). This review is structured into chapters: introduction, Glycosaminoglycans: Structure, biosynthesis, and catabolism, Intracellular organelle impairment, Modulation of altered intracellular pathways as a potential therapeutic approach, and Perspectives.
The review is readable and understandable and introduces the reader to the subject of MPS.
Unfortunately, there are a few inaccuracies or errors in the thesis that still need to be corrected:
- Lines 61-64 you wrote: “They are involved in line 62 critical biological processes, including ECM hydration, cell signaling, and regulation of line 63 growth factors (Table 1)” but in Table 1 there are only nemae of GAG, Monosaccharides constituents and chemical structure! It is the same with the description of Table 1: "Table 1. GAGs structure and their biological functions”. Please thoroughly review the text referencing the table and unify the text and content of the table.
Answer. We agree with the reviewer and have fixed it. Please see lines 61 and 140
- Line 159: please, move the description of Table 2 to a new page above the table.
Answer. We have followed this author’s suggestion. Please see lines 188-192
- Line 518 you wrote: ” As mentioned above (section 4.6)…” However, this review doesn't have section 4.6! Please carefully read this text and correct the links to the other sections.
Answer. We appreciate this comment and have changed 4.6 to 3.6 which is the section regarding immune response activation. It was a misspelling. Please see line 542.
- Finally, the weakest part I find section 5 Perspectives. Please try to improve this section with your findings and what would be key to focus research on in the near future. Or where you see the most potential in research.
Answer. We appreciate this comment and have improved that section. Please see lines 585-590.
Reviewer 2 Report
The review is quite comprehensive and informative. In the first part, the structure, biosynthesis, and catabolism of glycosaminoglycans are explained. In the second part, classification of MPS and major characteristics are specified. In the next sections, the authors explained the critical aspects of organelles impairment and potential interventions for their recovery . My only suggestion to the authors is to write the long name of DAMPs, which is abbreviated and briefly mentioned in Figure 1. Damage-associated molecular patterns have been important in understanding the molecular basis of many diseases in recent years. A few sentences can be added on this subject. The review is written in such a way as to be a book chapter in content. It will be a good resource for basic and clinical scientists working on Lysosomal Diseases.
Thanks for your hard work.
Author Response
Reviewer 2
Comments and Suggestions for Authors
The review is quite comprehensive and informative. In the first part, the structure, biosynthesis, and catabolism of glycosaminoglycans are explained. In the second part, classification of MPS and major characteristics are specified. In the next sections, the authors explained the critical aspects of organelles impairment and potential interventions for their recovery.
- My only suggestion to the authors is to write the long name of DAMPs, which is abbreviated and briefly mentioned in Figure 1. Damage-associated molecular patterns have been important in understanding the molecular basis of many diseases in recent years. A few sentences can be added on this subject.
Answer. We appreciate this kind suggestion and have included it accordingly. Please see line 294. Additionally, we agree with the reviewer about the novel understanding of DAMPs in many diseases including MPS. In this regard, we have included additional information. Please see lines 436-438.
The review is written in such a way as to be a book chapter in content. It will be a good resource for basic and clinical scientists working on Lysosomal Diseases.
Thank you very much for your positive comments.